# NLRP7 Promotes Choriocarcinoma Growth and Progression through the Establishment of an Immunosuppressive Microenvironment

**DOI:** 10.3390/cancers13122999

**Published:** 2021-06-15

**Authors:** Deborah Reynaud, Roland Abi Nahed, Nicolas Lemaitre, Pierre-Adrien Bolze, Wael Traboulsi, Frederic Sergent, Christophe Battail, Odile Filhol, Vincent Sapin, Houssine Boufettal, Pascale Hoffmann, Touria Aboussaouira, Padma Murthi, Rima Slim, Mohamed Benharouga, Nadia Alfaidy

**Affiliations:** 1Institut National de la Santé et de la Recherche Médicale, Inserm U1292, Université Grenoble-Alpes, 38000 Grenoble, France; deborah.reynaud.89@gmail.com (D.R.); rolandabinahed@gmail.com (R.A.N.); nicolas.lemaitre@cea.fr (N.L.); Frederic.sergent@cea.fr (F.S.); christophe.battail@cea.fr (C.B.); odile.filhol-cochet@cea.fr (O.F.); PHoffmann@chu-grenoble.fr (P.H.); mohamed.benharouga@cea.fr (M.B.); 2Commissariat à l’Energie Atomique et aux Energies Alternatives (CEA), Biosciences and Biotechnology Institute of Grenoble, CEDEX 9, 38054 Grenoble, France; 3Department of Gynecological Surgery and Oncology, Obstetrics, French Reference Center for Gestational Trophoblastic Diseases, University Hospital Lyon Sud, University of Lyon 1, 69000 Lyon, France; pierre-adrien.bolze@chu-lyon.fr; 4Laboratory for Immuno-Oncology, Lombardi Comprehensive Cancer Center, Georgetown University Medical Center, Washington, DC 2005, USA; wt247@georgetown.edu; 5Genetics, Reproduction and Development (GReD) Laboratory, CNRS UMR 6293, Inserm U1103, Translational Approach to Epithelial Injury and Repair Team, Clermont Auvergne University, 63000 Clermont-Ferrand, France; vincent.sapin@uca.fr; 6Medical Biochemistry and Molecular Biology Department, CHU Clermont-Ferrand, 63000 Clermont-Ferrand, France; 7Obstetrics and Gynecology Department, Ibn Rochd University Hospital, Hassan 2 University, Faculty of Medicine and Pharmacy, 20360 Casablanca, Morocco; mohcineb@yahoo.fr (H.B.); aboussaouira@gmail.com (T.A.); 8Centre Hospitalo-Universitaire Grenoble Alpes, Service Obstétrique, CS 10217, Université Grenoble Alpes, CEDEX 9, 38043 Grenoble, France; 9Department of Pharmacology, Monash Biomedicine Discovery Institute, Monash, Clayton, VIC 3168, Australia; padma.murthi@monash.edu; 10Department of Obstetrics and Gynecology, The University of Melbourne, Parkville, VIC 3010, Australia; 11Departments of Human Genetics and Obstetrics and Gynecology, McGill University Health Centre Research Institute, Montréal, QC H4A 3J1, Canada; rima.slim@muhc.mcgill.ca

**Keywords:** choriocarcinoma, hydatidiform mole, NLRP7, tumor microenvironment, maternal immune tolerance, orthotopic model of choriocarcinoma

## Abstract

**Simple Summary:**

*NLRP7* is the major gene responsible for recurrent hydatidiform mole (HM), an abnormal pregnancy that can develop into gestational choriocarcinoma (CC). Among women with one HM (sporadic HM), about 1–9% develop a second mole (recurrent HM). While the association of biallelic mutations in *NLRP7* with recurrent HM is well established, its role in the development and the immune tolerance of CC was unknown. The present work was conducted to investigate the direct involvement of NLRP7 in the development of CC and to provide evidences of its contribution in maternal immune tolerance. The study demonstrates that NLRP7 is directly involved in CC growth and downregulates the maternal immune response, thus fostering tumor growth and dissemination. Our study proposes that an increased expression of NLRP7 plays a significant role as a facilitator of CC growth, and therefore, NLRP7 should be categorized among important actors of CC development. The clinical relevance of NLRP7 in this rare female reproductive cancer highlights its therapeutic promise as a molecular target.

**Abstract:**

The inflammatory gene *NLRP7* is the major gene responsible for recurrent complete hydatidiform moles (CHM), an abnormal pregnancy that can develop into gestational choriocarcinoma (CC). However, the role of NLRP7 in the development and immune tolerance of CC has not been investigated. Three approaches were employed to define the role of NLRP7 in CC development: (i) a clinical study that analyzed human placenta and sera collected from women with normal pregnancies, CHM or CC; (ii) an in vitro study that investigated the impact of NLRP7 knockdown on tumor growth and organization; and (iii) an in vivo study that used two CC mouse models, including an orthotopic model. NLRP7 and circulating inflammatory cytokines were upregulated in tumor cells and in CHM and CC. In tumor cells, NLRP7 functions in an inflammasome-independent manner and promoted their proliferation and 3D organization. Gravid mice placentas injected with CC cells invalidated for *NLRP7*, exhibited higher maternal immune response, developed smaller tumors, and displayed less metastases. Our data characterized the critical role of NLRP7 in CC and provided evidence of its contribution to the development of an immunosuppressive maternal microenvironment that not only downregulates the maternal immune response but also fosters the growth and progression of CC.

## 1. Introduction

NLRP7 is a member of the NOD-like receptors (NLR), a family of proteins that plays a crucial role in the innate immune response. The NLRs are activated within the cell by pathogen-associated molecular patterns (PAMPs) or by damage-associated molecular patterns (DAMPs) [1,2], and their activation results in the formation of an inflammasome [3]. The inflammasome is defined as an intracellular multimeric protein complex that contains a sensor receptor (NLR) [3], an adaptor protein called ASC [3,4], and an effector enzyme, caspase-1. This complex catalyzes a cellular reaction that leads to the processing and maturation of two proinflammatory cytokines, the interleukin (IL)-1β and IL-18 [3,5]. Beside its proinflammatory role against immediate dangers [6], NLRP7 overexpression has been reported to function in an inflammasome-independent manner and exhibit anti-inflammatory role by inhibiting IL-1β production [7,8].

NLRP7 has extensively been studied in relation to the female reproductive system, as its recessive mutations are associated with recurrent hydatidiform moles (HM) [9]. The primary role of NRLP7 in recurrent HM is in the oocyte where NLRP7 associates with other proteins, and they altogether form a complex called the subcortical maternal complex (SCMC) that is specific to the mammalian oocyte and localizes at its cortex [10,11]. In humans and mice, biallelic mutations in several members of the SCMC are responsible for a spectrum of reproductive conditions that include infertility, early embryonic arrest during preimplantation development, recurrent miscarriages and HM, and multilocus imprinting disturbances [9,12,13]. CC is a malignant trophoblastic tumor that can develop after normal or abnormal pregnancies (live birth, complete (CHM), partial HM (PHM), miscarriages, and ectopic pregnancies) [14,15]. CHM develops when one or two spermatozoa fertilize an oocyte that either does not contain a nucleus or loses it after fertilization, while PHM results from dispermic fertilization of a nucleated oocyte [14,15]. Both CHM and PHM patients are at high risk of developing postmolar neoplasia; nevertheless, this risk is much higher after CHM (5–20%) than after PHM (2–3%) [16]. Patients with recessive *NLRP7* mutations have diploid biparental recurrent moles and are not at higher risk of developing CC than patients with sporadic HM (CHM or PHM), and their molar tissues, in general, have less trophoblastic proliferation than sporadic androgenetic CHM [8,17]. CC has an estimated incidence of 2–7 in 100,000 pregnancies in North America and Europe [16]. This incidence is higher in Africa [18,19] and Asia, where it may reach 5–202 in 100,000 pregnancies [20,21]. CC is an invasive cancer that may spread throughout the body with the most common sites of metastases being the lungs [22], liver, and brain [20,21].

One of the main features of CC is the excessive proliferation of the trophoblast, a phenomenon that ultimately results in an increased pool of cells acquiring migratory and invasive phenotype [23]. CC is also highly metastatic due to the intrinsic invasive property of the trophoblast [24]. We recently showed that *NLRP7* is highly expressed in the placenta during the first trimester of pregnancy and plays a critical role in the control of key developmental processes of the placenta. In normal placenta, NLRP7 functions as an inflammasome and increases trophoblast proliferation and controls its differentiation [25]. In relation to cancer, increased NLRP7 expression has been reported to be associated with poor prognosis of colorectal cancer [6], endometrial cancer [26], and to play a crucial role in testicular tumorigenesis [27]. However, no studies have investigated the mode of function of NLRP7 in CC and whether it contributes to CC development and aggressiveness.

Here, we show that NLRP7 is highly expressed in CHM, CC, and in a human CC-derived cell line, JEG3. We next investigated the effects of NLRP7 knockdown on JEG3 proliferation and organization using both 2D and 3D culture systems. We also conducted an in vivo study using our newly developed orthotropic animal model of CC to compare the effects of *NLRP7* knockdown on JEG3 development into CC tumors upon their injection in the placenta of gravid mice. To determine the contribution of the placenta to CC development, *NLRP7*-inactivated JEG3 cells were injected in the uterine horn of nongravid mice. Extensive analyses using immunohistochemistry, cytokine-array and RNA-seq were undertaken to decipher the mechanism by which NLRP7 contributes to CC development.

## 2. Materials and Methods

### 2.1. Human Study

#### 2.1.1. Normal and Pathological Human Tissue and Sera

Normal and pathological material consisting of placental tissues and sera from patients were collected from the French Reference Center for Gestational Trophoblastic Diseases using the national biobank for the study of gestational trophoblastic diseases [28]. Fixed and paraffin-embedded placental were also obtained from the McGill University Health Center Research Institute and from Ibn Rochd Hospital in Casablanca. We used the following sampling: CHM (*n* = 26), CC (*n* = 11) and normal placental tissues collected from normal first-trimester pregnancies (*n* = 29). Collection and processing of human tissues were approved by local hospital ethics committees, and informed patient consent was obtained in all cases. Appendix A summarizes all patient demography used in this study.

#### 2.1.2. Immunohistochemistry of the Human Placentas

Human placental tissues were collected from normal pregnant women during the first trimester of pregnancy and from patients with CHM or CC. Tissues were processed as described previously [29]. Human NLRP7 antibody was used at a final concentration of 13.6 µg/mL (Covalab, Bron, France). We have also used the monoclonal pan-cytokeratin antbody to characterize trophoblast cells in CHM and CC human tissues (BioLegend, San Diego, CA, USA, AE1/AE3 clone).

#### 2.1.3. ELISA

The concentrations of inflammatory cytokines IL-1β, IL-6, IL-8, IL-10, and TNFα were performed using a commercially available ELISA obtained from PeproTech (PeproTech, Neuilly-sur-Seine, France). IL-18 ELISA kit was purchased from R&D, (France). Levels of cytokines were quantified in the patient sera according to the manufacturer’s protocols. The levels of the cytokines measured were in a linear range of a standard curve as previously described [30].

### 2.2. Cell Culture

#### 2.2.1. HTR8/SV Neo Cell Line Culture

The human normal extravillous trophoblast cell line, HTR8/SV neo (ATCC^®^ CRL3271™) (referred hereafter as HTR8 for simplicity), was derived from the first-trimester chorionic villi explants transfected with the simian virus 40 large T antigen gene. These cells were used between 24 and 30 passages and grown in RPMI-1640 medium supplemented with 10% fetal bovine serum (FBS), penicillin-streptomycin (100U/mL) and amphotericin B (0.25 µg/mL) (Invitrogen, Cergy Pontoise, France) and maintained in a humidified 37 °C, 5% CO_2_ incubator.

#### 2.2.2. JEG3 Cell Line Culture

JEG3 (ATCC^®^ HTB-36™) is a CC cell line model. JEG3 cells were cultured in DMEM: F12, 10% FBS and maintained in a humidified 37 °C, 5% CO_2_ incubator. They were regularly tested for mycoplasma contamination and used for experimentation when the cells were between 4 and 10 passages.

#### 2.2.3. JEG3 Luc Cell Line Preparation

JEG3-Luc (Luciferase positive JEG3) [31], were prepared using a lentivirus supernatant (pLenti-II-CMV-Luc-IRES-GFP control vector). The protocol was performed according to the manufacturer instructions (Applied Biological Materials Inc., Richmond, BC, Canada). In brief, JEG3 cells were plated in DMEM: F12 (1/1) medium supplemented with 10% FBS and infected with lentivirus at a ratio of 1:1, for 6 h. The cells were then grown in selective medium with Neomycin (200 μg/mL) for 7 days.

#### 2.2.4. JEG3-Sh-NLRP7 Cell Line Preparation

JEG3-Sh-NLRP7 (NLRP7 knockdown JEG3-Luc) were prepared using a lentivirus supernatant prepared from bacteria transformed by Sh-NLRP7 and Sh-CTL plasmids (MISSION pLKO.1-puro; Sigma-Aldrich, St. Louis, MO, USA). Five ShRNA (1–5) against *NLRP7* were developed: Sh1 (TRCN0000128829); Sh2 (TRCN0000149236); Sh3 (TRCN0000148811); Sh4 (TRCN0000128007); and Sh5 (TRCN0000148388). TRCN stands for The RNA Consortium (TRC) guidelines. The protocol was performed according to the manufacturer’s instructions (Sigma-Aldrich). JEG3 cells were infected for 4 h with the lentivirus at a ratio of 1:2 in complete medium supplemented with polybrene (8 µg/mL, Sigma-Aldrich). Infected cells were selected with puromycin (10 μg/mL) for 7 days.

### 2.3. 2D Culture System

#### Proliferation Assay

JEG3-Sh-CTL and JEG3-Sh-NLRP7 cells were compared for their proliferation rate using [^3^H]-thymidine incorporation. They were cultured overnight (7 × 10^4^ cells/well, in a humidified 37 °C, 5% CO_2_ incubator), then incubated for 24 h with 0.5 µCi/mL [^3^H]-thymidine (Amersham, France), and subsequently washed in HBSS and incubated in 2 mL ice-cold 5% trichloroacetic acid for 20 min at room temperature. After washing, NaOH (0.1 M) and SDS (0.1%) were added, the lysates were transferred into a vial containing scintillation liquid, and the radioactivity was counted in a beta counter for CPM (Beckman, Krefeld, Germany) [32,33,34].

### 2.4. 3D Culture System

#### 2.4.1. Anchorage-Independent Spheroids Formation

Confluent monolayers of JEG3-Sh-CTL and JEG3-Sh1-NLRP7 cells were trypsinized. In total, 1500 cells from each of the cell types were suspended in DMEM: F12 supplemented with 10% FBS and seeded onto nonadherent round-bottom 96-well plates (Greiner, Pleidelsheim, Germany) precoated with Poly-HEMA (Poly 2-hydroxyethyl methacrylate), (Sigma Aldrich). Under these conditions, all suspended cells contributed to the formation of a single JEG3 cell spheroid specimen. The kinetic of the growth of each of the spheroids was assessed using a time-lapse imaging system (Incucyte, EssenBioscience, Ann Arbor, MI, USA). Quantification of the spheroid growth of JEG3-Sh-CTL and JEG3-Sh1-NLRP7 were analyzed using ImageJ software after 7 days of culture. At least six replicates for each of the cell types were performed within each experiment.

#### 2.4.2. Anchorage-Independent Colony Formation Assay in Soft Agar

Agar base containing 0.6% (w/v) of Noble agar (Becton Dickinson, Franklin Lakes, NJ, USA) dissolved in complete JEG3 growth medium was plated in 12-well plate and dried for 1 h at room temperature. Next, 30,000 JEG3-Sh-CTL and JEG3-Sh1-NLRP7 were suspended in 0.3% (w/v) of soft agar in complete growth medium and plated on the top layer of the base agar. After 14 days of growth, colonies were stained with 10% Coomassie Brilliant Blue (R250 staining solution, Biorad Laboratories, Inc., Hercules, CA, USA) overnight at 4 °C. Images were taken by Zeiss AxioVision microscope and processed using AxioVision SE64 Rel. 4.9.1 software. ImageJ software was used to determine the total number and size of the colonies.

### 2.5. RNA Isolation and Real-Time PCR Analysis

Total placental RNA was extracted from CHM and age-matched control placental tissues using the Trizol reagent (Invitrogen, Carlsbad, CA, USA) as previously described [35]. Total RNA from HTR8 and JEG3 cells were extracted using Macherey Nagel RNA extraction kit according to the manufacturer protocol. Reverse transcription was performed on total RNA (1 µg) (Iscript, Biorad). The primers used for the RTq-PCR are reported in Appendix A. The mRNA expression of the target genes was quantified using the real-time RT-qPCR on a Bio-Rad CFX96 apparatus using the GoTaq qPCR Master Mix (Promega, Madison, WI, USA). PCR conditions were as follows: step 1, 94 °C for 10 min; step 2, 35 cycles consisting of 95 °C for 15 s; step 3, 60 °C for 5 s and 72 °C for 10 s. Furthermore, the relative quantification of mRNA expression of the target genes was performed using the comparative threshold (CT) method by determining the CT values for the reference and the target genes in each of the sample sets according to the E−∆∆Ct method. Changes in mRNA expression level were calculated following normalization to the endogenous control genes RPl0, 18S rRNA, and GAPDH.

### 2.6. Western Blot Analysis

Total protein extracts were prepared as previously described [34]. Human NLRP7 antibody was used at 1.3 µg/µL (Covalab France), anti-IL-1β at 1µg/mL (Santa Cruz, Dallas, TX, USA), and PCNA at 0.1 µg/mL (Becton Dickinson). Immunoreactivity was detected using chemiluminescence detection kit reagents and a Chemidoc Station (Bio-Rad, Berkeley, CA, USA). To standardize for sample loading, the blots were subsequently stripped using a commercially available stripping kit, following the manufacturer’s instructions (Reblot mild solution, Millipore) and reprobed for the endogenous control protein using a mouse anti-β-actin antibody (Sigma-Aldrich), which was used as an internal control for total protein loading [29].

### 2.7. Gene Sequencing

Sequencing of *IL-1β* and *NLRP7* genes in HTR8 and JEG3 cells used in this study was performed by genomic DNA amplification of all their exons by PCR and Sanger sequencing in both directions. Sequences were aligned with the following reference sequences for *IL-1β* and *NLRP7*; NM_000576 and NM_001127255.1, respectively, and analyzed [36].

### 2.8. In Vivo Studies

#### 2.8.1. Experimental Groups

All animal studies were approved by the institutional guidelines and those formulated by the European Community for the Use of Experimental Animals (APAFIS#8596-2017011911162643). Mice were grouped according to the three different protocols, which was categorized based on the site of injection and delivery of JEG3-luc. In group 1, protocol-1, mice were injected, and JEG3-luc was delivered directly into their placenta; in protocol-2, they were injected and delivered into the uterine horns. For protocol-1, two–three months old SHO SCID female mice were mated in the animal facility. The presence of a vaginal plug was observed at 0.5 day postcoitum (dpc). The gravid mice were randomly assigned to be injected by JEG3-luc-CTL or JEG3-Sh-NLRP7 cells (see flowchart for detailed protocol, Appendix A). At least seven animals were assigned in each of the groups of the mice in which injection and delivery was performed in the placentas. These mice were injected at 7.5 dpc in two opposed placentas with JEG3-Sh-CTL (*n* = 8) and JEG3-Sh-NLRP7 (*n* = 7). Nongravid mice were injected and delivered in their uterine horns with JEG3-Sh-CTL (*n* = 3) and JEG3-Sh-NLRP7 (*n* = 3).

#### 2.8.2. Histology and Immunohistochemistry of Mouse Tissues

Mouse tissues were processed as previously described [37]. For immunohistochemistry, the following tissues, placentas and uterine horn, were stained for the following antibodies; anti-Ki67, clone MIB1 antibody, used at 0.61µg/mL (Dako, Carpinteria, CA, USA); anti-hCG, ready to use antibody (Dako); anti-PD-L1 (Cell signaling); anti-HLA-G clone 4H84 (Covalab); anti-F4/80 clone BM8 (eBioscience, San Diego, CA, USA). The use of the beta-hCG form was preferred to the hyperglycosylated-hCG (H-hCG) because forming syncytiotrophoblast from JEG3 cells mainly expresses this hormone [38].

#### 2.8.3. Bioluminescence Imaging

Mice imaging was performed at days 14.5 dpc and at day 19.5 dpc for the gravid mice. Mice injected in the uterine horns were imaged 7 and 12 days following the injections. Bioluminescence imaging was performed with a highly sensitive, cooled CCD camera, mounted in a light-tight specimen box (IVIS^®^, In Vivo Imaging System, PerkinElmer, Waltham, MA, USA). Fifteen minutes before imaging, animals were anesthetized with 2% isoflurane and injected with luciferin (potassium salt, Xenogen, Alameda, CA, USA; 10 µL/10g of body weight). This dose and route of administration have been shown to be optimal for studies in rodents when images were acquired within 15 min postluciferin administration.

For imaging, mice were placed onto the warmed stage inside the light-tight camera box, with continuous exposure to isoflurane. The data were acquired for 45 s for all mice. The low levels of light emitted from the bioluminescent tumors were detected by the IVIS^®^ camera system and were then integrated, digitized, and displayed. The regions of interest (ROI) from displayed images were designated around the tumor area and were quantified as total photon/s, using Living Image^®^ software (Xenogen, Alameda, CA, USA).

After imaging of the whole body of the mouse, a laparotomy was performed to collect blood and to expose and image the uterine horn containing embryos with their attached placentas, as well as the rest of metastatic organs. A second imaging of the organs was performed and quantified as described above. Placentas and metastatic organs were collected and stored at −80 °C for Western blot and RNAseq analyses or fixed in PFA (4%) for immunohistochemistry. Blood was collected to measure cytokines using ELISA and antibody-arrays as described earlier.

### 2.9. RNA-Seq Analysis

The RNA sequencing (RNA-seq) library preparation was performed with the SMART-Seq v4 ultralow input RNA kit (Clontech, Mountain View, CA, USA) by GENEWIZ (Leipzig, Germany). Libraries were constructed using the Illumina Nextera XT kit and analyzed for concentration (Qubit DNA assay and NanoDrop; Thermo Fisher Scientific, Waltham, MA, USA), size distribution (Agilent Bioanalyzer, Beijing, China), and quantification of viable sequencing templates via qPCR. Sequencing was performed on the Illumina HiSeq in Rapid Run Mode with 2 × 150 bp single-end configuration. Sequencing reads were assessed for overall quality, followed by adapter trimming and removal of low-quality data. They were then mapped to the mouse reference genome using CLC Genomics Workbench (QIAGEN) followed by quantification of gene expression in hit counts and in RPKM values. Differential gene expression analysis was performed using DESeq2 (Bioconductor). Genes associated with an adjusted *p* value < 0.05 and an absolute log2 fold change >1.5 were considered to be significantly differentially expressed. Biological pathway enrichments were performed by gene set enrichment analysis (GSEA, FDR < 0.25) using biological processes (BP) annotation from Gene Onlogy GO.db_v3.10.0 (Bioconductor R3.6.3) database. GSEA was executed by ClusterProfiler v3.14.3 (Bioconducture R3.6.3). Volcano plot was generated by EnhancedVolcano_v1.4.0 (Bioconductor R3.6.3).

### 2.10. Antibody Cytokine Arrays

Antibody arrays were used to compare the expression profile and levels of inflammatory-related cytokine (62 targets) in sera collected from the groups of mice injected in the placenta. Antibody array experiment was assessed using mouse angiogenesis array kit (Abcam, Cambridge, UK) as recommended by the manufacturer. The intensities of immunoreactive bands were analyzed using the Chemidoc analyzing system (Image Lab Version 4.0.1) and measured by scanning the photographic film and analyzing the images using the ImageJ software.

### 2.11. Statistical Analysis

Statistical comparisons were made using Mann–Whitney, Student’s *t*-test, and one-way ANOVA. All data were checked for normality and equal variance. Where normality failed, a nonparametric test followed by Dunn’s or Bonferroni’s test was used. (SigmaPlot and SigmaStat, Jandel Scientific Software, San Jose, CA, USA). All data expressed as means SEM (standard error to the mean) and considered significant when the *p* value is < 0.001, 0.01, or 0.05.

## 3. Results

### 3.1. NLRP7 Inflammasome Is Highly Expressed in CHM and CC

We previously showed that *NLRP7* is highly expressed in the human placenta and that its levels are elevated during the first trimester of pregnancy [25]. Here, we compared *NLRP7* levels in placental tissues obtained from women with normal pregnancies (control) to those of obtained from CHM and CC using immunohistochemistry. We found that NLRP7 protein was predominantly localized to the syncytiotrophoblast layer and was highest in CC followed by CHM and then normal placenta, Figure 1A,B. Tumor trophoblast cells were characterized using pan-cytokeratin staining, Appendix A. We next investigated the levels of transcripts of NLRP7 transcripts and those of IL-1β and IL-18, the two cytokines that reflect inflammasome activities. We found a significantly higher amount of *NLRP7*, *IL-1β*, and *IL-18* transcripts in CHM as compared to control placental tissues, Figure 1C–E.

To gain more insight into the status of the NLRP7 inflammasome effectors in patients with CHM and CC, we compared the circulating levels of IL-1β and IL-18, Appendix A. Again, we found significant increases in the levels of IL-1β in the sera of patients with CHM and CC, with the highest levels being in patients with CC followed by those in patients with CHM, then control subjects. IL-18 levels were significantly decreased in patients with CHM patients and unchanged in CC patients compared to age-matched sera from control women. Further comparisons of other key inflammatory cytokines, such as IL-6, IL-8, TNFα, and the anti-inflammatory cytokines, IL-10, showed that these cytokines are all increased in the sera of women with CHM compared to those of control women, Appendix A.

### 3.2. NLRP7 Inflammasome Is Not Active in the Human Choriocarcinoma Cell Line, JEG3

To further investigate the role of NLRP7 in CC, we used the human choriocarcinoma cell line, JEG3. First, we compared the transcriptional levels of *NLRP7* and its effectors, *IL-1β* and *IL-18*, in JEG3 and normal (nontumorigenic) trophoblast cells, HTR8. We found that *NLRP7* mRNA levels were higher in JEG3 than in HTR8 cells, Figure 1F. IL-1β mRNA was only detected in HTR8 cells but not in JEG3 cells, Figure 1G, while IL-18 was detected at comparable levels in both JEG3 and HTR8 cells, Figure 1H. At the protein level, NLRP7 expression was significantly higher in JEG3 as compared to HTR8 cells. The absence of expression of IL-1β in JEG3 cells was also confirmed at the protein level in the cellular lysates, Figure 1I, and in the culture supernatants, Figure 1J. To investigate the causes of the lack of IL-1β transcripts and proteins in JEG3 cells, we PCR amplified and sequenced all the 6 exons of IL-1β (including the 5′ UTR) on JEG3 genomic DNA but did not identify any mutation (data not shown). We also sequenced the 11 exons of NLRP7 on JEG3 genomic DNA and again did not identify any mutation (data not shown). Therefore, the lack of IL-1β transcripts and proteins in JEG3 cells is not due to mutations in its coding or regulatory regions but to other intrinsic properties of JEG3 cells. Altogether, these findings demonstrate that IL-1β, the main effector of NLRP7 inflammasome, is not active in JEG3 cells.

### 3.3. NLRP7 Knockdown in JEG3 Cells

To characterize further the role of *NLRP7* in JEG3 cells, we downregulated its expression using short hairpin (Sh) strategy. Comparison of NLRP7 protein expression in the five Sh-NLRP7 and Sh-CTL showed that the Sh1 exhibited the most significant decrease in *NLRP7* protein level, Appendix A. Hence, Sh1-NLRP7 was used in all subsequent in vitro and in vivo experiments. We then compared the levels of *NLRP7* protein in JEG3-Sh1-NLRP7 and JEG3-Sh-CTL. This analysis demonstrated that JEG3-Sh1-NLRP7 exhibited significant decrease in *NLRP7* mRNA, Appendix A, and protein levels, Appendix A.

### 3.4. NLRP7 Knockdown Decreases the Proliferation of JEG3 Tumor Formation In Vitro

One of the key features of gestational choriocarcinoma is their high proliferation. To determine whether NLRP7 is involved in this process, we compared the proliferative potential of JEG3-Sh-CTL and JEG3-Sh-NLRP7. Figure 2A depicts a comparison of [^3^H]-thymidine incorporation between the two cell types following 24 h of culture. JEG3-Sh-NLRP7 exhibited a significant decrease in their proliferation rate when compared to JEG3-Sh-CTL cells. These findings strongly suggest that NLRP7 is involved in the proliferation of choriocarcinoma trophoblastic cells.

It was particularly relevant to determine the effect of *NLRP7* knockdown on tumor formation by JEG3 cells in 3D culture systems, which form structures and topologies similar to the ones observed in vivo [37]. Using both the colony and spheroid formation techniques, we demonstrated that *NLRP7* knockdown decreased the initial processes of tumor formation. Figure 2B,C shows that JEG3-Sh-NLRP7 developed less colonies than JEG3-Sh-CTL, and the sizes of these colonies were significantly smaller in JEG3-Sh-NLRP7. This effect was substantiated through a follow-up study on the formation of spheroids by JEG3-Sh-NLRP7 and JEG3-Sh-CTL during 10 days of culture. Figure 2D shows representative photomicrographs of JEG3-Sh-CTL and JEG3-Sh-NLRP7 forming spheroids at day 3 and day 10. We also observed that JEG3-Sh-NLRP7 spheroids exhibited lower diameters and were not as organized as the JEG3-Sh-CTL spheroids. Using time-lapse microscopy, we further confirmed that JEG3-Sh-NLRP7 spheroids grew more slowly than JEG3-Sh-CTL spheroids. The difference was significant from day 5 as depicted in Figure 2E.

### 3.5. Role of NLRP7 in CC Development and Progression in Two Preclinical Mouse Models

The above observations using clinical samples and in vitro studies strongly suggested that NLRP7 might contribute to the development and growth of CC. To test this hypothesis, we used our newly developed orthotopic mouse CC model [37] that mimics CC development and progression in humans. Using this model, we compared tumor formation and growth by injecting JEG3-Sh-CTL or JEG3-Sh-NLRP7 in the placenta of gravid mice. To determine the placental contribution to choriocarcinoma cell’s dissemination, we also used a nongravid mice that was injected by the same cell number in the uterine horns. Appendix A reports the flowcharts of the two in vivo models.

### 3.6. NLRP7 Knockdown Decreases CC Tumor Growth in the Mouse Placenta

To mimic gestational CC development and progression from its primary site, the placenta, we injected JEG3-Sh-CTL or JEG3-Sh-NLRP7 cells orthotopically within the placenta of SHO-SCID mice at 7.5 dpc. Mice were then monitored for tumor development and metastasis until 19.5 dpc. Figure 3A shows that JEG3-Sh-CTL developed trophoblastic tumors outside the placenta, which is a key feature for diagnosing CC, as early as 7 days after injection (Figure 3Aa), while JEG3-Sh-NLRP7 cells did not show similar tumor growth outside the placenta (Figure 3Ac). At day 19.5 of gestation (12 days post injection), we observed an increase in the growth of CC cell masses in the JEG3-Sh-CTL condition (Figure 3Ab) and a stabilization of the growth of CC cells in mice injected with the JEG3-Sh-NLRP7 (Figure 3Ad). In addition, at 19.5 dpc, there was a decrease in the number of metastatic sites in mice injected with the JEG3-Sh-NLRP7, Figure 3Ae–f. Quantification of the tumor growth at 14.5 and 19.5 dpc is reported in Figure 3B–D.

Comparison of the levels of expression of the proliferation marker PCNA showed that placentas collected from JEG3-Sh-CTL mice exhibit higher expression of the PCNA proteins as compared to those collected from JEG3-Sh-NLRP7, Figure 3E (Appendix A shown original western blots). These data strongly suggest that in vivo the high expression of NLRP7 by JEG3 cells increases their proliferation and growth potential. In addition, gravid mice injected with JEG3-Sh-CTL had significantly more resorbed fetuses at 19.5dpc compared to mice injected with JEG3-Sh-NLRP7, Figure 4A.

Histological comparison of placentas collected from mice injected with JEG3-Sh-CTL or JEG3-Sh-NLRP7 showed that tumors developed in 80% of the placentas collected from JEG3-Sh-CTL group versus in only 40% of the placentas collected from JEG3-Sh-NLRP7 group. Placentas collected from mice injected with JEG3-Sh-CTL exhibited important histological changes with loss of all placental structures and zones compared to placentas collected from mice injected with JEG3-Sh-NLRP7, Figure 4B. To better characterize these tumors, we used the following antibodies, anti-Ki67 (clone MIB1) that stains only human proliferating cells; anti-hCG that stains human hCG expressing trophoblast cells and antibodies against human HLA-G and PD-L1 [39]; two membrane proteins expressed by human extravillous trophoblast and syncytiotrophoblast, respectively. In addition, HLA-G and PD-L1 play key roles in suppressing the immune system during pregnancy and tumorigenesis. To characterize the host immune response, we used anti-F4/80-Ab that stains mouse macrophages. Figure 4C–E show representative photomicrographs of this staining. We observed less proliferating (Figure 4C) and hCG-expressing cells in JEG3-Sh-NLRP7 placentas. In addition, tumor cells in the JEG3-Sh-NLRP7 group expressed less HLA-G and PD-L1 (Figure 4D). In both groups, F4/80 staining showed that mouse macrophages are adjacent to the tumors but did not infiltrate them (Figure 4E). Altogether, these data suggested that high *NLRP7* expression by CC cells increases the expression of key markers that inhibit the host immune response and could enable these cancer cells to escape, proliferate, and disseminate.

### 3.7. NLRP7 Knockdown Decreases CC Tumor Growth in the Uterine Horn

To determine the contribution of the placenta to CC development, JEG3-Sh-NLRP7 cells were injected in the uterine horn of nongravid mice and analyzed tumor formation 7 and 12 dpi. As for the injection in the placenta, we observed that *NLRP7* knockdown in JEG3 cells significantly reduced tumor development and growth, Figure 5A. Quantifications of tumor growth at 7 and 12 dpi are reported in Figure 5B,C, respectively. Importantly, we observed that at 7 dpi, the tumors are smaller upon JEG3-Sh-NLRP7 or JEG3-Sh-CTL injection in the uterine horn as compared to the injections of the same cells in the placenta Figure 5B,C. These data suggest that a placental environment promotes the development of CC.

Histological comparison of uteri collected from the two groups showed that mice injected with JEG3-Sh-CTL exhibited larger tumor masses than those injected with JEG3-Sh-NLRP7, which had smaller and necrotic tumors, Figure 6A. To better characterize these tumors, we used the same antibodies used to characterize the placentas. Figure 6B,C reports representative photomicrographs of the staining. Similar to the results obtained on placental tumors, less proliferating and hCG-secreting cells were observed in JEG3-Sh-NLRP7 tumors. Again, tumor cells in the JEG3-Sh-NLRP7 condition expressed less HLA-G and PD-L1, and mouse macrophages appeared to slightly infiltrate the JEG3-Sh-NLRP7 tumors but not the JEG3-Sh-CTL tumors. These data suggest that JEG3-Sh-NLRP7 tumors created an immunocompetent environment that negatively controlled their growth.

### 3.8. NLRP7 Knockdown Rescued Maternal Adaptive Immune Response

To determine whether the tumor environment contributes to tumor growth, we compared the mouse transcriptome around placental tumors of eight mice injected with JEG3-Sh-CTL and seven mice injected with JEG3-Sh-NLRP7 using RNA-seq analysis. Differential mouse gene expression analysis using DEseq2 method between the environment of JEG3-Sh-CTL and JEG3-Sh-NLRP7 tumors identified 21 genes that were differentially expressed (adjusted *p*-value < 0.05 and absolute log2 fold change >1.5), 20 genes were upregulated, and 1 gene was downregulated. These 21 differentially expressed genes are reported in a volcano plot that shows for each gene (indicated by red dots) the logarithm of its adjusted *p* value as a function of the logarithm of its fold change of expression between JEG3-Sh-NLRP7 and JEG3-Sh-CTL tumor environments, Figure 7A. Biological pathway enrichment analysis using GSEA (gene set enrichment analysis) method between JEG3-Sh-NLRP7 and JEG3-Sh-CTL tumor environments highlighted the activation of the immune system-related processes in JEG3-Sh-NLRP7 tumor environment. The activation of the adaptive immune response pathway in the JEG3-Sh-NLRP7 tumor environment is further illustrated by its GSEA enrichment plot, Figure 7B. These processes included the adaptive immune response, positive regulation of the immune system, and regulation of leucocytes migration and differentiation. The GSEA analysis also identified suppressed pathways that are associated with the regulation of stem cell population maintenance and trophoblast differentiation in the JEG3-Sh-NLRP7 tumor environment, Figure 7C.

We next used an antibody array to analyze a selection of circulating inflammatory factors in JEG3-Sh-NLRP7 and JEG3-Sh-CTL-injected mice in the orthotopic model, Figure 7D and Appendix A. Densitometry analysis showed that five cytokines, IL-4, AXL, IL-1β, IL-9, Exotaxine-2 and IL2, were upregulated, and two cytokines, PF4 and IL-6, were downregulated in the tumor environment of JEG3-Sh-NLRP7 cells. Importantly, most of the upregulated cytokines are associated with immune cell priming and induce antitumor response in the tumor microenvironment. These cytokines were also reported to be involved in the innate and adaptive immunity [40,41,42,43,44,45]. In contrast, most of the downregulated cytokines were reported to induce tumor development [46,47,48,49]. The levels of some of these circulating proteins are used in FIGO (International Federation of Gyneacology and Obstetrics) histology scores to estimate gestational trophoblastic tumor size, prognosis of patients and to predict the overall mortality.

### 3.9. A Proposed Model for NLRP7 Contribution to the Development of CC In Vivo

Taken together, the above results lead us to the working model about the role of NLRP7 in gestational CC, Figure 8. The top panel shows a representative CC cell with an elevated cytoplasmic expression of *NLRP7* that is associated with an increase in the surface expression of hCG as well as of PD-L1 and HLA-G, two proteins known to inhibit the functions of several immune cells, such as natural killer (NK), and might render HLA-G and PD-L1 positive cells resistant to immune rejection. This phenotype may confer these cells the ability to evade from the maternal immune system, which may facilitate further their growth and metastasis to other organs. The lower panel shows the same tumor cell but invalidated for *NLRP7*. The loss of NLRP7 expression is associated with a decrease in the levels of expression of hCG, HLA-G, and PD-L1. This phenotype prevents tumor cells from hiding from the maternal immune cells, which then surround the tumor. The activation of the maternal immune system leads to an increase in the release of cytokines such as, IL-1β, IL-9, IL-17, Eotaxin-2 and IL2 in the tumor environment and subsequently in the maternal circulation. The loss of the camouflage of the JEG3 cell was substantiated by the increase in the tumor microenvironment of the expression of genes directly involved in chemotaxis and migration of immune cells including macrophages and monocytes.

## 4. Discussion

The present work demonstrates the direct involvement of NLRP7 in the development of CC and provides the evidence of its contribution to the development of an immunosuppressive microenvironment that fosters tumor growth and progression. NLRP7 is the major gene responsible for recurrent CHM, a benign tumor of the placenta that may develop into CC in 5–20% of cases [16]. However, no study has yet investigated NLRP7 involvement in the etiology of this cancer. Through the present study, we propose that NLRP7 is a key actor of CC growth that should be categorized among the important pathophysiological factors for the development of this cancer.

These statements are based on three major observations. First, NLRP7 expression is higher in the trophoblast of CHM and CC as compared to normal trophoblast cells of matching gestational stages. These results are of significant clinical importance as NLRP7 can be used as a prognostic biomarker of placental cancer development and/or potentially as a new target for CC treatment [50]. These findings are in line with the recently reported role of NLRP7 in the control of normal trophoblast proliferation [25,36,51]. Second, the strong expression of NLRP7 contributes to the survival of CC cells, increases their proliferation rate, and facilitates their 3D organization. Third, the two animal models used in this study substantiate the clinical and in vitro results and further support our findings demonstrating that NLRP7 overexpressing cells create an environment that downregulates the host immune response, which contributes further to the growth and dissemination of tumor cells.

The role of NLRP7 has only been reported for reproductive cancers of the testis and the endometrium [26]. In testicular cancer, NLRP7 has been shown to play a role in cell proliferation [27]. In the endometrium, NLRP7 expression was associated with poor patient prognosis [26]. The association of NLRP7 with CC development is not surprising since NLRP7 exhibits all features of a placental protein that controls key developmental aspects of human placentation, such as trophoblast proliferation and differentiation and is the major mutated gene in recurrent CHM [25,36,51].

The demonstration that NLRP7 is highly expressed in CC suggested that its inflammasome is also activated; however, we did not observe IL-1β production or secretion by JEG3 cells, suggesting that NLRP7 role in JEG3 cells is independent of its inflammasome machinery. This assumption is in line with previous reports demonstrating that NLRP7 overexpression in vitro may exert a negative feedback regulation on IL-1β production [7,8]. Importantly, numerous studies reported that IL-1β negatively controls trophoblast proliferation by affecting directly the cell cycle [52,53,54]. In line with these findings, Chow et al. demonstrated that NLRP3 promotes metastasis independently of its inflammasome activity and that *Nlrp3^−/−^* mice have lower numbers of lung metastases after intravenous inoculation of melanoma or prostate carcinoma cells [55]. Recent studies also reported that an increased *NLRP12* expression is associated with the progression of prostate cancer in the absence of increased levels of mature IL-1β or IL-18 by cancer cells [56]. Altogether, these findings strongly suggest that NLRP7, similar to NLRP3 and NLRP12, may function in an inflammasome-independent manner in cancer [56,57].

The association of high *NLRP7* expression with the development of CC in vivo was established using the newly developed animal model [37], via the injection of CC cells within the placenta and confirmed with another route, the uterine horn. The placental injection route emphasized the role of this highly vascularized organ in the growth of these tumor cells since mice injected in the uterine horn did not exhibit the same degree of tumor growth and dissemination.

From a clinical perspective, the data obtained in vivo may explain what may occur in women with CHM who go on to develop CC. In fact, numerous studies proposed that CC would often develop in women with a weak immune system, which confers a tolerant environment for tumor development [8,58]. The SCID mouse model used in our study mimics the weak immune system of CC patients as these mice do not have mature B and T lymphocytes, but they have normal natural killer cells, macrophages, and granulocytes [59]. In relation to the NK role in NLR-related cancer, it was reported that resistance to metastasis in the *Nlrp3*^−/−^ mice was fully attributed to enhanced NK-cell activity [55]. More importantly, this study demonstrated that NLRP7 knockdown in CC tumor cells was associated with a decrease in the expression of numerous proteins involved in maternal immune tolerance such as HLA-G, PDL1, and hCG. Importantly, the group of *A. Schumacher* demonstrated that hCG increases the number and activity of regulatory T cells (Treg) and retains tolerogenic dendritic cells (DCs) [60,61]. Hence, these results further supports a local immune tolerance of secreted hCG by tumor cells that acts as chemoattractant for T-suppressors (T-Treg) and apoptotic actors for T-lymphocytes.

## 5. Conclusions

In conclusion, we demonstrated that *NLRP7* is highly expressed in CHM and CC patients; it functions in an inflammasome-independent manner in CC cells and contributes to CC development, both in vitro and in vivo. Furthermore, we show that NLRP7 plays another important role in tumor development, through its influence on tumor cells environment. Altogether, these studies provide strong evidences of the involvement of high *NLRP7* expression in the development of gestational choriocarcinoma. The clinical relevance of NLRP7 in this rare female reproductive cancer highlights its potential therapeutic promise as a molecular target to treat choriocarcinoma.

## Figures and Tables

**Figure 1 cancers-13-02999-f001:**
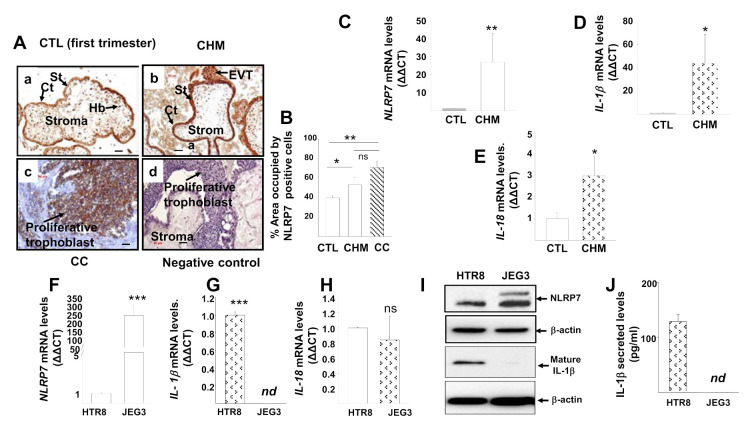
NLRP7 expression in the placenta of Control, CHM, and CC patients during the first trimester of pregnancy. (**A**) reports representative photomicrographs of NLRP7 immunoreactivity in placental villous tissues from CTL (**a**), CHM (**b**), and CC (**c**) patients. Photomicrograph in (**d**) is a negative control. Cytotrophoblast (Ct), Hofbauer cells (Hb), extravillous trophoblast (EVT), and syncytiotrophoblast (St). Scale bar = 50 μm. (**B**) depicts comparisons of levels of NLRP7 protein expression in CTL, CHM, and CC placental sections, * *p* < 0.05, ** *p* < 0.01 ± SEM. (**C**–**E**) report comparisons of *NLRP7*, *IL-1β* and *IL-18* mRNA levels in placentas of CTL (*n* = 7) and CHM (*n* = 11) women during the first trimester of pregnancy, respectively * *p* < 0.05, ** *p* < 0.01 ± SEM. (**F**–**H**) report comparisons of *NLRP7*, *IL-1β*, and *IL-18* mRNA levels in HTR8 and JEG3 cells, *** *p* < 0.001 ± SEM. (**I**) depicts comparisons of NLRP7 and mature IL-1*β* protein levels in HTR8 and JEG3 cells. Standardization of protein signals was performed using antibodies against β-actin. (**J**) reports comparison of IL-1*β* secreted levels in HTR8 and JEG3 cells. ns: not significant; nd: not detectable.

**Figure 2 cancers-13-02999-f002:**
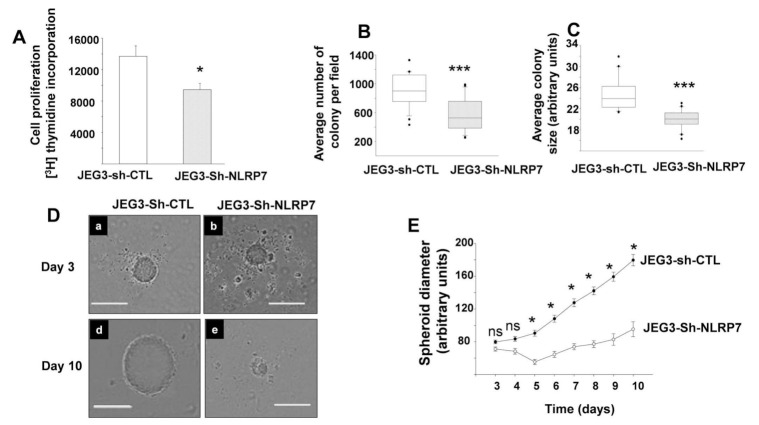
Comparison of JEG3-Sh-CTL and JEG3-Sh-NLRP7 proliferation and colony and spheroid formation in 3D culture systems. (**A**) reports the effect of *NLRP7* knockdown on the proliferation JEG3 cells. Cell proliferation was determined using [^3^H] thymidine incorporation. Data are presented as a mean ± SEM * *p* < 0.05. (**B**) reports the quantification of the number of colony formation by JEG3-Sh-CTL and JEG3-Sh-NLRP7 per field. Data are presented as a mean ± SEM. *** *p* < 0.001. (**C**) depicts a graph that compares the average sizes of the colonies formed by JEG3-Sh-CTL and JEG3-Sh-NLRP7. Data are presented as a mean ± SEM. *** *p* < 0.001. (**D**) shows representative images that compare JEG3-Sh-CTL and JEG3-Sh-NLRP7 spheroid formation at day 3 after their seeding (**a**,**b**), and 10 days post culture (**d**,**e**). The scale bar is 400 µm. (**E**) reports kinetics that compares changes in the spheroid diameter of JEG3-Sh-CTL and JEG3-Sh-NLRP7 from day 3 to day 10 of culture. (*n* = 3 experiments in hexaplicats) * *p* < 0.05. Data are presented as a mean ± SEM.

**Figure 3 cancers-13-02999-f003:**
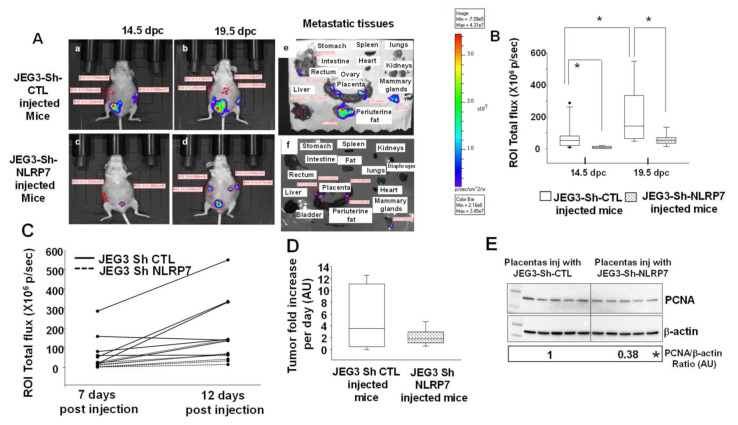
Comparison of tumor growth in gravid mice injected with JEG3-Sh-CTL or JEG3-Sh-NLRP7 in the placenta. (**A**) shows representative luminescent images of gravid mice injected by JEG3-Sh-CTL (**Aa**,**Ab**) or JEG3-Sh-NLRP7 (**Ac**,**Ad**). (**Aa**,**Ac**) images were taken at 14.5 dpc (7 days post injection, dpi), and images (**Ab**,**Ad**) were taken at 19.5 dpc (12 dpi). Photomicrographs in (**Ae**,**Af**) show images of metastatic organs. (**B**) reports quantification of the values of photon flux (p/sec) for JEG3-Sh-CTL-injected mice (*n* = 8) and JEG3-Sh-NLRP7 (*n* = 7) at 14.5 and 19.5 dpc, respectively. Data are presented as box plots, * *p* < 0.05. (**C**) illustrates tumor growth in JEG3-Sh-CTL and in JEG3-Sh-NLRP7 mice from the 7th to 12th dpi. (**D**) reports the comparisons of the tumor fold increases in the JEG3-Sh-CTL (*n* = 8) and in JEG3-Sh-NLRP7 (*n* = 7) groups. (**E**) shows Western blot analysis and quantification of the PCNA protein levels in the placenta collected from JEG3-Sh-CTL and in JEG3-Sh-NLRP7 mice. Standardization of immunoreactivity was performed using antibodies against β-actin. Data are standardized to control. * *p* < 0.05.

**Figure 4 cancers-13-02999-f004:**
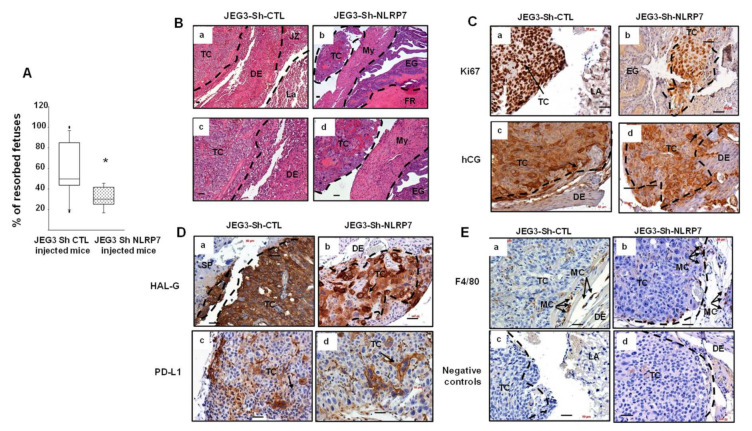
Comparison of tumor growth in gravid mice injected in the placenta and immunohistological analyses. (**A**) compares the percentages of resorbed fetuses in JEG3-Sh-CTL and in JEG3-Sh-NLRP7 mice at day 19.5 of gestation. * *p* < 0.05. Data are presented as a mean ± SEM. (**B**) depicts representative images of histological comparison of placenta collected from mice injected with JEG3-Sh-CTL (**Ba**,**Bc**) or JEG3-Sh-NLRP7 (**Bb**,**Bd**). Photomicrographs in (**b**,**d**) are higher magnification of those reported in (**a**,**c**), respectively. TC: tumor cells, DE: decidua; La: labyrinth, JZ: junctional zone; My: myometrium; EG: endometrial glands; and FR: fetal resorption. Scale bar = 50 µm. (**C**–**E**) report representative images of placental sections collected from JEG3-Sh-CTL and JEG3-Sh-NLRP7-injected mice and stained with different antibodies. (**Ca**,**Cb**) show Ki67 staining; (**Cc**,**Cd**) hCG; (**Da**,**Db**) HLA-G; (**Dc**,**Dd**) PD-L1; (**Ea**,**Eb**) F4/80; and (**Ec**,**Ed**) represent negative controls that were incubated without the primary antibodies. TC: tumor cells, EG: endometrial Gland; SP: spongiotrophblast; DE: decidua; LA: labyrinth, and MC: macrophages. Scale bar = 50 µm.

**Figure 5 cancers-13-02999-f005:**
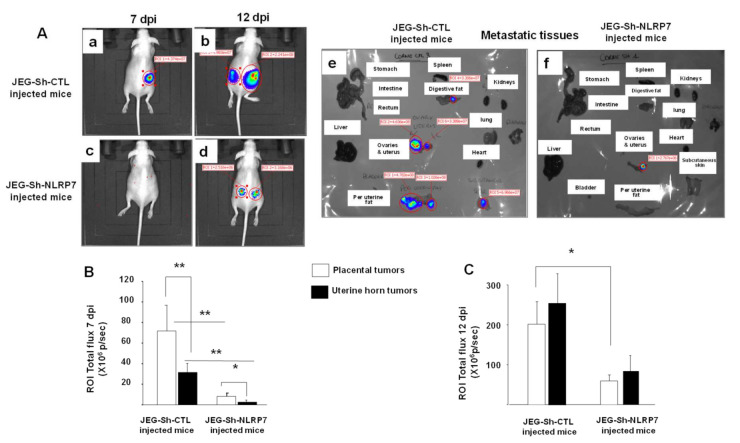
Comparison of tumor growth in nongravid mice injected in the uterine horns with JEG3-Sh-CTL or JEG3-Sh-NLRP7. (**A**) shows representative images of female mice injected by JEG3-Sh-CTL (**Aa**,**Ab**) or JEG3-Sh-NLRP7 (**Ac**,**Ad**) at day 7 and day 12 dpi, respectively. Images in (**Ae**,**Af**) show images of metastatic organs (**B**,**C**) report quantification and comparison of the values of photon flux (p/sec) for mice injected in the uterine horn or in the placenta (recalled data) with JEG3-Sh-CTL to those injected with JEG3-Sh-NLRP7 at 7 and 12 dpi. Data are presented as mean *±* SEM. * *p* < 0.05, ** *p* < 0.01 ± SEM.

**Figure 6 cancers-13-02999-f006:**
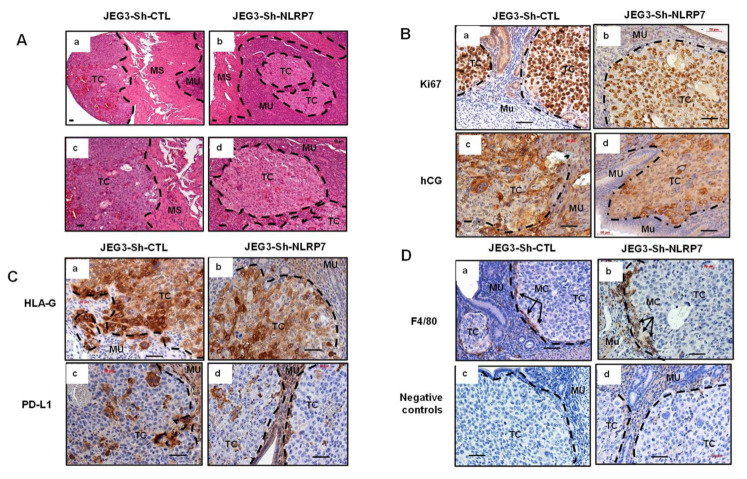
Immunohistochemical analyses of tumors collected from mice injected in the uterine horns with JEG3-Sh-CTL or JEG3-Sh-NLRP7. (**A**–**D**) show representative images of the uterine horn sections collected from JEG3-Sh-CTL and JEG3-Sh-NLRP7-injected mice. (**Aa**,**Ab**) depicts histology of the uterine horns; (**Ac**,**Ad**) are higher magnification of the upper microphotographs; the subsequent section shows immunoreactivity for the following proteins. (**Ba**,**Bb**) show Ki67 staining; (**Bc**,**Bd**) hCG; (**Ca**,**Cb**) HLA-G; (**Cc**,**Cd**) PD-L1; (**Da**,**Db**) F4/80; (**Dc**,**Dd**) represent negative controls that were incubated without the primary antibodies. TC: tumor cells, MU: mucosa; MS: muscularis, MC: macrophages. Scale bar = 50 µm.

**Figure 7 cancers-13-02999-f007:**
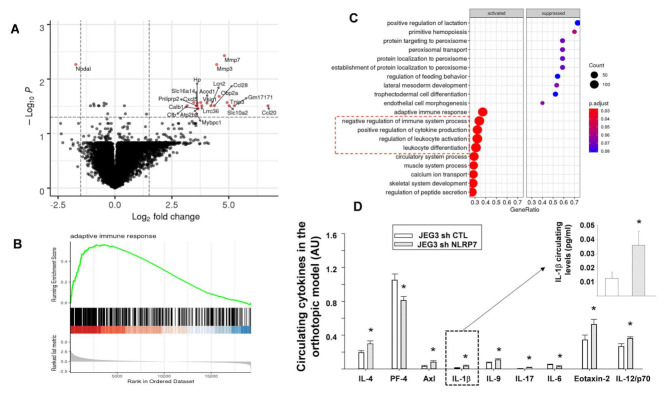
Effects of *NLRP7* knockdown on placental gene expression and circulating cytokines. (**A**) shows a volcano plot that shows for each gene (indicated by dots) the logarithm of the fold change between expression levels in JEG3-Sh-NLRP7 versus JEG3-Sh-CTL tumors according to the logarithm of the adjusted *p*-value generated from differential gene expression analysis. Red dots depict significantly over- or under-expressed genes (adjusted *p*-value < 0.05 and absolute log2 fold change >1.5). (**B**) reports the GSEA enrichment plot of the adaptive immune response pathway. The top portion shows the running enrichment score (ES) for the gene set as the analysis walks down the ranked gene list. The score at the peak of the plot is the ES for the gene set. A positive ES indicates gene set enrichment at the top of the ranked gene list (i.e., genes overexpressed in JEG3-Sh-NLRP7). The middle portion shows where the members of the gene set appear in the ranked list of genes. The bottom portion shows the value as the logarithm of the fold change, between expression levels in JEG3-Sh-NLRP7 versus JEG3-Sh-CTL cells, used as gene ranking metrics. (**C**) reports a dot plot that represents the molecular processes identified as the most significantly enriched by the GSEA (gene set enrichment analysis) method. The top 10 activated or suppressed pathways were selected based on their enrichment in under- or over-expressed genes in JEG3-Sh-NLRP7, respectively. The gene ratio corresponds to the proportion of genes deregulated by pathway in relation to the total number of genes annotated by pathway. The red dashed box highlights key pathways of immune adaptive responses. (**D**) shows a selection of circulating proteins that were significantly different in the antibody array analyses of 62 cytokines, between JEG3-Sh-CTL (*n* = 6) and JEG3-Sh-NLRP7 (*n* = 4) placenta-injected mice. Data are represented as mean ± SEM. * *p* < 0.05.

**Figure 8 cancers-13-02999-f008:**
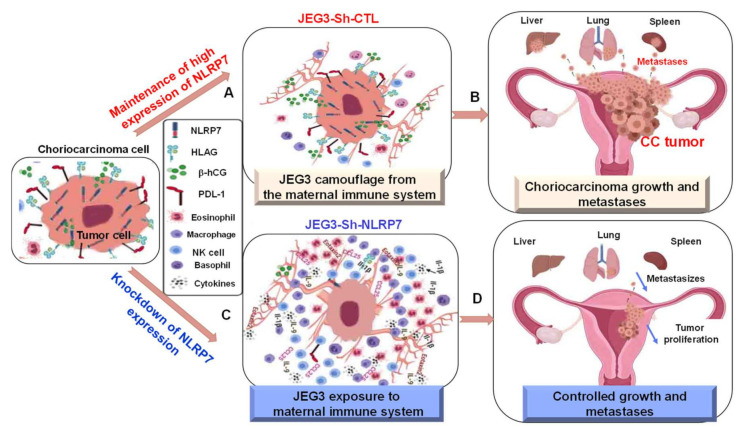
Proposed model for the NLRP7 contribution into the development of CC in vivo. The model shows on the left a representative CC tumor cell within its environment. To note that CC cell highly expresses, NLRP7, PD-L1, HLA-G and β-hCG. (**A**) shows that this phenotype confers a camouflage to this tumor cell from the maternal immune system, facilitating its growth in a tolerant immune environment. This leads to choriocarcinoma tumor to grow within the placenta and metastasize toward other organs, such as the lung, the liver, and spleen, (**B**). (**C**) shows the same tumor cell but invalidated for NLRP7. The reduced NLRP7 expression causes a decrease in the levels of PD-L1, HLA-G, and β-hCG expression. This phenotype caused a failure in the camouflage process, which triggers the maternal immune system. The activation of this system caused an increase in the release of local cytokines, such as IL-1β, IL-9, and Eotaxin. This leads to a control in the growth and metastasis of choriocarcinoma tumor (**D**).

## Data Availability

All data in our study will be available upon reasonable request.

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
