# Peer review of "NLRP7 Promotes Choriocarcinoma Growth and Progression through the Establishment of an Immunosuppressive Microenvironment"

_cancers, 2021, doi:10.3390/cancers13122999_

Round 1

Reviewer 1 Report

The authors previously described that NLRP7 functions as an inflammasome and plays an important role in trophoblastic cell proliferation and differentiation.  In this study, they investigated its roles in malignant trophoblastic cells and on tumor development. They first described high expression of NLRP7 in the placenta of CHM and CC patients (Is it possible to distinguish mutated and wt NLRP7 on these samples?) and demonstrated that wt NLRP7 (Is NLRP7 mutated in JEG3 cells?) could contribute to CC development both in vitro and in vivo. In addition, they suggested that it could play an important role in tumor development by influencing tumor microenvironment.

It is known that NLRP7 mutations are associated with HM. Are these forms more stable than wt NLRP7? Are they also biologically active? It is an important aspect to determine if NRLP7 could play a role in the degeneration of HM in CC.

I have also comments on the manuscript.

1/ In simple summary, could you explain what is recurrent HM?

2/ Material and methods

  • 2.1.1: How many samples of control, CHM and CC from each center did you use? And is it possible to obtain patients characteristics ?
  • 2.5: one reference is missing (noted as REF)

3/ Results

  • figure 1: 7 CTL and 11 CHM were immunolabeled. How these samples were chosen (which biobank)? Hofbauer cells were noted Hb in the figure and Ho in the legend. It should be great to add immunostaining for trophoblastic cells characterization (HLA-G, CK7).
  • 3.2: Five ShRNA were developed. Could you indicate what are the numbers starting with TRCN please? This information should be included in Mat and Met and not in results section.
  • 3.8: FIGUREO: did you want to indicate IFGO?

Author Response

Please find response to reviewer a as attched file.

Reviewer 2 Report

Reynaud and colleagues have shown that NLRP7 ablation helps the immune system against choriocarcinoma. The work contains novel findings that clearly show the involvement of NLRP7 in choriocarcinoma progression.

I have a couple of suggestions listed below:

Introduction

The introduction is well written and covers the majority of needed information for the rest of the manuscript. However, authors can also discuss the role of NLRP7 as a member of Subcortical maternal complex (SCMC). NLRP7 mutations are responsible for biparental complete hydatidiform mole (BiCHM). However, no mutations are identified in NLRP7 in association to androgenetic moles, unexplained infertility and recurrent pregnancy loss. Authors can easily find the related literature.

Results and discussion

Results and discussion are very detailed and precise, however, there is no introduction about hCG. HCG interferes with immune system by affecting various immune cell populations like B cells, dendritic cells and T cells. It attracts regulatory T cells into the fetal maternal interface and orchestrates immune tolerance toward the fetus. Interestingly low levels of Treg cells in spontaneous abortion patients is associated with low amounts of hCG. I would suggest to add some information in this regard. Also, few words about the variated hCG – Hyperglycosylated form -in choriocarcinoma and invasive mole may help to understand better the reason for which authors checked hCG.

Author Response

Please find attached the response to reviewer 2

Round 2

Reviewer 1 Report

The authors answered to the comments of reviewers and modified their masnucript appropriately.